# Review of Ireland’s First Year of the COVID-19 Pandemic Impact on People Affected by Eating Disorders: ‘Behind Every Screen There Was a Family Supporting a Person with an Eating Disorder’

**DOI:** 10.3390/jcm10153385

**Published:** 2021-07-30

**Authors:** Harriet Parsons, Barry Murphy, Deirbhile Malone, Ingrid Holme

**Affiliations:** 1Bodywhys, The Eating Disorders Association of Ireland, P.O. Box 105, Blackrock, Co Dublin, Ireland; communications@bodywhys.ie (B.M.); deirbhile.n.malone@gmail.com (D.M.); 2School of Sociology, University College Dublin, Dublin, Ireland; ingrid.holme@ucd.ie

**Keywords:** eating disorders, COVID-19, pandemic, parents, carers, online support groups, helpline, family support programme

## Abstract

Irish society went into one of the most stringent lockdowns in March 2020 due to the COVID-19 pandemic, and barring a few weeks, remains highly restricted at time of writing. This has produced a wide range of challenges for those affected by eating disorders, as well as treatment services and Bodywhys, The Eating Disorders Association of Ireland. Current research indicates that COVID-19 has impacted across three key areas—the experience of those with an eating disorder, the experience of service provision, and the impact on the family situation. Drawing on monitoring and evaluation data gathered by Bodywhys, this paper details the challenges faced by those affected by eating disorders in Ireland and how the organisation responded to these challenges, providing support in various forms to people with eating disorders and their families.

## 1. Introduction

Initially, Ireland responded quickly and decisively to the COVID-19 pandemic. The first national case was confirmed on 29 February 2020. On March 11 the World Health Orgnisation (WHO) declared the Coronavirus (COVID-19) outbreak a pandemic, and the first Irish fatality was recorded. A day later, on 12 March 2020, the Government announced the closure of schools, colleges, and childcare facilities [1]. In a public address on 17 March 2020, the Taoiseach (Prime Minister), Leo Varadkar, asked people to ‘come together as a nation by staying apart from each other’ [2]. On 24 March 2020, the Taoiseach announced stringent new measures designed to halt the spread of COVID-19, describing them as ‘unprecedented actions’ to respond to an ‘unprecedented emergency’. Currently, Ireland has been under one of the longest COVID-19 lockdowns in Europe with restrictions in place since Christmas 2020. Over the past year, restrictions have been adapted many times (see Table A1 in Appendix A for timepoints), leading to uncertainty both for individuals and organisations. This paper reviews how people affected by eating disorders in Ireland were impacted by the COVID-19 pandemic, drawing on monitoring and evaluation data gathered by Bodywhys, the Eating Disorders Association of Ireland, which works alongside public and private treatment providers.

The Irish health system is divided into the public service available to all and the private service available to those with private health insurance. The public health service is provided by the Health Service Executive (HSE). Within the HSE’s mental health section, there are services provided at different levels according to assessed risk. Primary care, community mental health teams, day patient and inpatient services. Specifically, for the treatment of people with eating disorders, in the main, treatment is provided in the community, by the Child and Adolescent Mental Health Team (CAMHS) or the Adult Mental Health Team (AMHS). To address eating disorders specifically, the HSE developed a National Clinical Programme for Eating Disorders (NCP-ED), launching the NCP-ED’s Model of Care in 2018 [3]. This is essentially a blueprint for the provision of skilled, experienced and specialist eating disorder teams nationally. No matter where the person is located, they will have access to a specialist team for their treatment. Importantly, Bodywhys was included on the working group for the development of NCP-ED and is identified as the partner to support the implementation of NCP-ED.

### 1.1. Impact of COVID-19 on People Affected by Eating Disorders in Ireland

There was a 66% increase in hospital admissions for eating disorders amongst children and adolescents in Ireland since the COVID-19 pandemic, compared to 2019 [4] and paediatric hospitals and acute hospitals have also reported increased acute presentations [5,6]. In Crumlin’s Children Hospital, Dublin, both community and hospital presentations for eating disorders increased during 2020, by three to four times, compared to 2019 [7]. Staff in Crumlin indicated that more acute presentations were evident, along with increases reported by their colleagues in Temple Street Hospital, Tallaght University Hospital and CAMHS. The HSE’s Child and Adolescent Regional Eating Disorder Service, CAREDS, for the Cork/Kerry region, reported two to three times the amount of young people needing help since the pandemic began [8]. In most of these examples, hospitalisations were underlined by the fact that people were more unwell, requiring stabilisation. Clinicians in Sligo/Leitrim Mental Health Services reported a 40% increase in eating disorder referrals amongst children and adolescents since March 2020 [9]. Increased inpatient emergency department admissions for eating disorders were reported in the Galway region [10]. In adult services, increased referrals, particularly in the 18–24 year old range, were also reported [11]. This includes people who have relapsed and those presenting with a new illness, higher instances of binge eating disorder and bulimia nervosa and increased referrals for males. A general practitioner (GP) working in a university student health setting, shared her experience of increased presentations for eating disorders, such as binge eating disorder and restrictive eating in an online news publication [12]. The HSE’s NCP-ED published its figures for 2020 during Eating Disorders Awareness Week (EDAW), 1–7 March 2021, noting a 60% increase in referrals received and a 43% increase in assessments completed, compared to 2019 [13]. Media reporting of the HSE’s mental health section indicates that its services expect increased eating disorder referrals to continue into 2021 [6].

Bodywhys, The Eating Disorders Association of Ireland, founded in 1995, is a national voluntary organisation with charitable status. Bodywhys was established when a group of parents, who had a child diagnosed with an eating disorder, joined together to form a support group. At that time, in Ireland, there were no specific support services for people affected by eating disorders. Primary funding for Bodywhys is provided by the HSE through a Service Level Agreement (SLA). This allows the organisation to deliver support services and expand other areas of work that benefit the public, such as prevention and education programmes, continuous professional development (CPD) training, media awareness and anti-stigma campaigns. Further details about the role of Bodywhys can be found in Appendix B.

### 1.2. Overview of Bodywhys Services

Bodywhys operates several support services for people affected by eating disorders. The helpline is a listening and information signposting service. Delivered by a team of trained volunteers, the helpline offers non-judgemental and confidential support and information. The helpline is open to anyone. A current limitation of the helpline is it has moved to two hours per day, four days per week. Previously, it was available through six weekly two hour slots. A strength of the helpline is that it is specifically for eating disorders and it is a space where people can share their lived experience. The email support service may be described as ‘like a helpline call, in an email’. Due to the large volume of support type requests being sent into the organisation’s administration email address, in 2005, Bodywhys developed a dedicated, free and confidential email support service for anyone based in the Republic of Ireland. This is open to anyone affected by eating disorders, including family, friends or partners, worried work colleagues, housemates, teachers, youth workers and other professionals. Prior to the pandemic, Bodywhys operated two types of face-to-face support groups: one for people with eating disorders and another for families and friends, in Dublin city centre. Historically, there were support groups in other regions in Ireland, but these were closed prior to the pandemic due to a variety of factors. The support group is a space where people can come together, with two trained facilitators to get and give support to one another. Bodywhys support groups are not therapy groups, although they often have a therapeutic effect, helping a person feel calmer and more able to cope. The Dublin support groups were suspended due to the pandemic however, the online support groups continued and grew in attendance. The online support groups, BodywhysConnect for adults aged 19 and over, and YouthConnect for young people aged 13–18, use text-based chat rather than video or audio communication. As with the face-to-face group, there are two trained facilitators at the group to ensure the group space is supportive and safe for all [14]. Apart from the cost of calling the helpline, all services provided by Bodywhys are free and self-referral in nature. They are open services, so people do not need to have a formal diagnosis to attend. The services provided by Bodywhys are independent of the traditional clinical space and do not interfere or interact with a person’s therapeutic alliance or treatment plan or services. Typically, people contact Bodywhys for support, information and understanding and support requests broadly fall into three categories: needing a listening ear, information or a person is at crisis point.

Another aspect of Bodywhys’ support services, is the free PiLaR family support programme. In 2014, Bodywhys piloted a structured programme for family members. This was developed into the PiLaR family support programme. Based on the idea of ‘Peer Led Resilience’, the family support programme brings families together for four evenings of learning, thinking, questioning and supporting. The programme provides psycho-education with skills, practical advice, with support. Covering topics such as, understanding an eating disorder as a coping mechanism, reframing behaviours to focus on the positive and learning, managing mealtimes, communication skills, learning about empathy and praise, recovery and treatment, coping with relapse, family members are supported to think about the support they are providing and how they are providing that. The PiLaR programme is delivered by the Bodywhys Training and Development Manager, and it is a strong illustration of the benefit of the collaborative and complementary relationship between Bodywhys and the HSE’s NCP-ED. In 2019, an evaluation of the programme was published by University College Dublin’s (UCD) School of Medicine and the HSE [15]. Clinicians alert families to the PiLaR programme for support and knowledge, assured that the information and support is consistent and not in conflict with what they are doing within the treatment setting. People can also register for the programme via the Bodywhys website, or by email. The programme enhances family readiness for treatment support and facilitates change. The programme is not for clinicians or for a person with an eating disorder.

Since the onset of the pandemic, Bodywhys staff have been working from home to deliver services and support remotely. In 2019, over 70% of contacts and queries on support services, excluding the PiLaR programme, came through an online source. Operating, adapting and transitioning to remote delivery, due to the pandemic, did not pose a significant challenge for the organisation due its significant experience in providing online services. All services continued to be provided, apart from the face-to-face support groups. The PiLaR family support programme went online. Bodywhys increased its services, setting up a regular support group for families who had attended PiLaR once they had finished the programme. This ‘post-PiLaR support group’ is only open to those who have completed the PiLaR programme and is limited to twenty people.

## 2. Methodology

This paper uses an organisational review approach using data from two sources.

### 2.1. Salesforce

Service numbers were extracted from Salesforce by the second author. Salesforce is a customer relationship management (CRM) cloud-based tool. Bodywhys records service user contact demographic details on its support services using customised log sheet templates on Salesforce. (Salesforce Inc., San Francisco, CA, United States). These log sheets for the support groups, online support groups, helpline and email contacts include, if disclosed, details such as location of contact (county), gender of contact, type of eating disorder, duration of eating disorder, if the query is from someone with an eating disorder or a relative, age demographics, if the person is in treatment, reason for the call—information, listening, or crisis. At registration for Bodywhys online support groups, users self-select options for age, gender and type of eating disorder. If a user accesses the online group, anonymous attendance figures are recorded noting age, gender, type of eating disorder and new or repeat user. Since 2009, Bodywhys staff have used Salesforce to create anonymous reports based on activities the organisation’s support services. Reports can be generated by selecting a support service and running an analysis on the previously described demographic variables and within specified date ranges. This information informs the organisation’s annual reports, and it is also provided to the HSE, including the HSE’s ‘psychosocial dashboard’ which has been in effect since the onset of the pandemic.

### 2.2. Organisational Service Monitoring and Evaluation Data

Evaluation and feedback surveys developed by the last author were compiled by Bodywhys were distributed among those who participated in the PiLaR programme at two distinct timepoints; Prior to Week One and Post Week Four. Data was collected from the Week One surveys which was completed from September 2020 onwards. These surveys obtained demographic information such as diagnosis received, relationship to the person with the eating disorder, and specific questions that aim to understand the effectiveness of conducting the PiLaR programme through an online platform. Alongside this, the following four questionnaires were included: Brief Resilience Scale [16], Fear of COVID-19 Scale [17] with two supplementary items focusing on eating disorders, Accommodation and Enabling Scale [18] and the Eating Disorders Symptom Impact Scale [19]. Ethics approval for collecting and processing these data fall under an ethics exception (HS-E-20-131-Holme).

From September 2020 to March 2021, 629 participants of the PiLaR programme submitted their responses to a survey compiled by Bodywhys (115 men, 510 women, 4 unspecified; *M_age_* = 47.20, *SD_age_* = 10.27, range = 14–70 years). The data of these participants were analysed using version 25 SPSS software platform computer package (IBM, Armonk, NY, United States) by the third author. The level of completion for each section varied and therefore, the data provided reflects the number of participants who completed that specific section (e.g., 585 participants completed the questions which measured the, ‘Impact on Mental Health of Carer’).

## 3. Results

### 3.1. Impact of COVID-19 Pandemic on Bodywhys Support Services

There has been an increasing demand for all Bodywhys services compared to the previous year: online support groups (111% increase, March–December 2019 vs. March–December 2020), support email contacts (45% increase, March–December 2019 vs. March–December 2020) and helpline calls (54% increase, April–December 2019 vs. April–December 2020). Figure 1 shows the participation levels across various services from January 2020 to February 2021.

One of the most significant changes for the organisation was taking the PiLaR programme online. Typically, the programme is delivered across Ireland, face-to-face, by Bodywhys staff in a conference room, hotel, parish hall or local community setting. In January 2020, PiLaR was delivered in Cork and Wicklow. On 4 March 2020, the programme started, in person, in Dublin. Due to the pandemic, social distancing requirements, and restrictions that were implemented in Ireland, the final two weeks of this programme moved online. The remainder of PiLaR programmes for 2020—May, September and November were delivered entirely online, through Zoom. Whilst different to providing the programme in person, it quickly became evident that behind every screen there was a family supporting a person with an eating disorder. Not surprisingly, the new delivery format meant there was significant growth in attendance. In total, 683 people attended in 2020, compared to 309 who availed of the programme 2019, reflecting an increase of 121% (PiLaR monitoring and evaluation data). According to the data, PiLaR attendees are supporting people aged 9 to 48 years old (46 men, 565 women, 1 non-binary, 1 transgender; *M_age_* = 18.32, *SD_age_* = 5.87).

A core concern for Bodywhys as an organisation is that the format of the interaction is accessible, safe and meets the participants’ expectations. Since 2019 Bodywhys has gathered data to ensure that its online technologies were easy to use. Analysis of PiLaR data indicates that 57% reported that it would not be difficult for them to find time to engage and 58% agreed that Zoom would be easy to use. A total of 61% of participants were comfortable with the level of online privacy while 57% reported that they were not concerned with being overheard while attending. However, according to the responses, 41% would have preferred the PiLaR programme delivered in a face-to-face format.

### 3.2. Impact of COVID-19 for Parents and Carers in Ireland

COVID-19 had a major impact on the general population, as well as those supporting people with ED in Ireland. The results from the ‘Fear of COVID-19 Scale’ (see Figure 2) indicated that there has been a consistently low level of general anxiety in relation to COVID-19. This has fluctuated to some degree (September 2020 (*M* = 9.3, *SD* = 5.2), November 2020 (*M* = 8.1, *SD* = 5.3), January 2021 (*M* = 9.3, *SD* = 5.2), February 2021 (*M* = 10.3, *SD* = 6.6), March 2021 (*M* = 9.4, *SD* = 6.5).

The degree to which carers felt that COVID-19 was impacting on their ability to care was also examined (see Figure 3). From September 2020, 34% of carers reported that COVID-19 was impacting their ability to provide support to the individual experiencing an eating disorder. This increased to 41% in November 2020 and 43% in January 2021. In February 2021, although there was a decrease in those who responded that they ‘agreed’ that COVID-19 was having an impact on their role as a carer, there was an increase in those who indicated that they ‘strongly agreed’ with this statement, resulting in a total of 51% of participants reporting that COVID-19 was impacting their ability to support the person with the eating disorder. This reflected the data reported in the, ‘Fear of COVID-19 Scale’ which indicated that participants experienced the highest COVID-19 related anxiety in February 2021 which consequently, may have impacted their ability to provide care for their loved ones. In March 2021, this figure decreased to 43%.

61% of the participants reported that their own mental health was being impacted by supporting someone with an eating disorder (see Figure 4). No significant change was identified in these figures over the course of seven months. See chart below.

Qualitative responses to open-ended questions in the survey also indicated that supporters were clearly felt worried about how to help:
*‘*I’m worried for the person in question because of COVID-19 and level 5 lockdown. And I was unsure how to help the person’.(PiLaR participant)


Alongside the difficulties that people with eating disorders were experiencing due to COVID-19, family members, friends and carers voiced the challenges of travel restrictions being in place.

‘Due to COVID-19 level 5 restrictions, I’m the only family member in the same county as my sister’.(PiLaR participant)

Results from the ‘Accommodation and Emerging Scale for Eating Disorders’ indicated that in February 2021 (*M* = 32.9, *SD* = 26.7), participants reported the highest figure in relation to the accommodations made by the family to support the individual experiencing an eating disorder (see Figure 5). This decreased in March 2021 (*M* = 30.3, *SD* = 23.1). According to the responses provided by the participants, the scores on the AESED were also higher in September 2020 (*M* = 31.7, *SD* = 23.1) before decreasing in November 2020 (*M* = 30.3, *SD* = 22.1) and further in January 2021 (*M* = 30.0, *SD* = 19.5).

These findings echo the anecdotal evidence gained from other Bodywhys services.

### 3.3. Impact of COVID-19 on People with Eating Disorders, as Reported to Bodywhys Support Services

Bodywhys does not gather routine, direct data from people experiencing eating disorders. However, insight is provided by examining the family reported data collected pre-attendance at PiLaR. The EDSIS scores suggest that the severity of eating disorder-related behaviours does fluctuate for those seeking to attend PiLaR (September 2020 (*M* = 32.48, *SD* = 19.0), November 2020 (*M* = 10.3, *SD* = 6.6), January 2021 (*M* = 9.3, *SD* = 5.2), February 2021 (*M* = 33.4, *SD* = 22.91), March 2021 (*M* = 29.6, *SD* = 16.74).

Bodywhys service users provided the organisation with anecdotal evidence of their fears and anxieties connected with all aspects of the eating disorder. People reported an intensification of eating disorder symptomatology, such as ‘eating disorder thoughts’, a reaction to the stress that had entered their lives, the sudden change to their routine, a sense of being ‘out of control’ and the threat to their recovery process. Common fears about recovery intensified. People with eating disorders mentioned fears about letting people down, about lapsing or relapsing, disappointing people, or wellness being judged by physical appearance. Many service users engaged in supports as a form of self-care, fearing relapse. Those who were doing well in recovery feared relapse due to the changed situation and stress and uncertainty that accompanied it. Many feared a reactivation of symptoms despite treatment. Evidence suggests a worsening of eating disorder symptoms and reduced coping abilities during the pandemic [20], whilst loneliness, increased interpersonal stress, balancing conflict priorities [21,22,23], household argument and fear about the safety of loved ones [24] have also been noted. A qualitative analysis of social media posts on the popular platform, Reddit, identified challenges including: disordered eating behaviours, negative body image, struggles with appetite and binge eating, loss of motivation to exercise, the stockpiling of food, anger, frustration and guilt, barriers to treatment, feeling ashamed of the associated health outcomes and a reluctance to tell others of physical distress or seek medical care [25].

In the early stages of the lockdown, fears of food shortages, shortages of ‘safe’ foods were issues, something seen as underscoring potential risk factors [26]. Anxiety from a sense of fear of being recognised, reactions to food purchases and feelings of shame when shopping has also been a worry [27]. People in contact with Bodywhys experienced an increased compulsion to engage in eating disorder behaviours, such as strict food restriction, exercise, binging and purging. The sense of being confined and an uncertainty about when the lockdown would end brought an increased sense of being out of control which in turn increased compensatory exercise behaviours, exacerbated binge eating in many cases, and increased purging behaviours. This disruption to routine and perceived control also triggered more rumination about weight, exercise and meals. Bodywhys began to hear of stress and anxiety triggering, what people called ‘emotional eating’. People reported their relationship with exercise challenging to manage. A need for secrecy about symptoms and hiding the eating disorder from others became an intense focus and fear for some. Recent literature points to emotions and disruption to routine as being amongst some the primary stressors for those affected by anorexia nervosa, bulimia nervosa and binge eating disorder [28]. Other challenges highlighted in this study were exposure to triggering messaging, changes in physical activity, changes in food availability and disruption to living situation, which also impacted by those affected by other specified feeding or eating disorder (OSFED).

Increased time online meant increased exposure to food and exercise messaging in public discourse, increased social comparisons, thereby exacerbating eating disorder thoughts. People reported that general health messages about needing exercise for mental wellness and physical fitness made them feel anxious and ‘not good enough’. In some cases, the health messages were taken to the extreme and set as excessive goals to reach. People with eating disorders tuned into the public sentiment that this confinement would cause weight gain, the so-called ‘quarantine 15’, (15 lbs body weight), evidenced by constant media messaging. People reported finding it very difficult to tune these messages out. One of the first eating disorder papers published during the pandemic identified eating disorder specific risk factors across a number of areas; including food access, exercise difficulties, media consumption and media messaging [26]. Reports from Bodywhys service users made it clear that the pandemic exacerbated communication difficulties. Video calls, a necessary new way of connecting with others, heightened awareness of the physical self, leading to self-criticism that is potentially harmful to recovery.

Limited access to healthcare [26], disruption to treatment outcomes [24], disparity in access to eating disorder services and premature discharge from services [29] and reduced contact from clinical teams [30] have been emerged as key concerns during COVID-19. These issues, along with reliance on remote support, premature discharge, disrupted transitions into community living, treatment suspensions, limited post-diagnostic support, remaining on waiting list for treatment all presented challenges and obstacles to those with an eating disorder since March 2020. Reduced time for professionals to address changes in treatment, more self-management of the illness by people with eating disorders were experienced as challenging not just for the person with an eating disorder but also for their carers. People already in treatment experienced uncertainty about how it would progress. Telehealth, medical checks and clinicians being reassigned to other healthcare jobs were issues. For those who were undergoing inpatient treatment, restrictions were imposed, resulting in minimum physical visits with family.

Since March 2020, Irish health services have reported increased medical admissions for anorexia nervosa, and anecdotally clinicians are seeing more people presenting, far more physically unwell and in an urgent condition [4,5,6,7,8,9,10,11,12]. Data from the Health Research Board (HRB) in Ireland indicates a 32% increase in adult hospital admissions for eating disoders, growing from 138 in 2019 to 182 in 2020 [31]. The 20–24 year olds were the most affected, though 18–44 year olds were significantly represented. For children and adolescents, figures increased from 54 in 2019 to 87 in 2020, a rise of 61%. This includes some repeat admissions. Reporting and media coverage has highlighted increased contact with eating disorders services has also occurred internationally. In Australia, hospital admissions have increased for anorexia nervosa, with patients requiring medical stabilisation, with factors such as anxiety and stress thought to play a role [32,33]. The Butterfly Foundation reported 150% increase in calls during the first school term compared to the same period 2019 [34]. In New Zealand, Auckland’s Starship Children’s Hospital, admissions of 10 to 15-year-olds doubled (from 33 in 2019 to 66 in 2020), along with admissions of under 20-year-olds at Auckland City Hospital [35]. A similar increase was seen across all ages at Waikato Hospital, while hospitals in Wellington saw a rise of 31%. In Canada, increased referrals have been documented in Ontario, for both inpatient and outpatient services [36,37]. In France, the Fédération Française Anorexie Boulimie (FFAB), reported a 30% increase in calls to its helpline in 2020 [38]. In the United States, the National Eating Disorders Association (NEDA) reported a 41% increase in messages to its telephone and online help lines in January 2021, compared with January 2020 [39]. In England, the number of children and young people seeking emergency support for anorexia nervosa and bulimia nervosa in the community reached an all-time high of 625 [40]. There were 19,562 new referrals of under-18s with eating disorders to NHS-funded secondary mental health services in 2020, a rise of 46% from the 13,421 new referrals in 2019 [41]. In London, between January and March 2021 there was a 40% increase in children receiving treatment, compared to January–March 2020 [42]. Support organisation Beat reported an increase of 173% in helpline calls between February 2020 and January 2021 [41] and a 120% increase in calls from Northern Ireland in 2020, compared to 2019 [43]. Previous research about an eating disorders helpline noted that those affected by anorexia nervosa and bulimia nervosa were interested in counselling, whilst those affected by binge eating disorder sought emotional support [44]. Eating disorder helplines can also serve as a first step, in parallel to treatment, or as an extension of this support when in recovery and helplines may provide an outlet for carers to access information and discuss their own experiences, while supporting their family member [45]. A recent evaluation carried out on behalf of support organisation, Beat, in the United Kingdom, found that its helpline helped service users understand their eating disorder better, reduced feelings of isolation and prompted them to act, such as contacting a doctor or read more information [46].

The experience of family members and carers also changed during the pandemic. Research thus far indicates that this was evident across three key areas—the experience of service provision, an impact on the family situation and carers’ observations of the health of people with eating disorders [27,47,48,49,50]. Specific concerns included a fear of premature discharge from services and changes in delivery of supports, increased signs of anxiety, new food triggers, signs of relapse, extra time at home and increased cooking frequency by people with eating disorders as compulsive and consuming. Managing multiple needs—people with eating disorders and family during lockdown, curtailed normal activities and lack of routine, challenges of shielding/cocooning, social distancing, feeling isolated as a carer and a lack of true understanding from others. Heightened unknown factors, creating meal plans at home, the support role of carers has increased due to stay at home requirements, adapting and coping with the needs of people with eating disorders and the wider family, whilst carers were also potentially now working from home or suddenly unemployed. Having to create new routines for children and keep them occupied and social contact with peers and school have also been issues managed by carers. Carers reported noticing that their family member who is unwell is withdrawn, irritable, depressive and engages in restrictive eating. The Maudsley Centre for Child and Adolescent Eating Disorders, which transitioned to online treatment delivery during COVID-19, found that parents (*n* = 19) generally felt more comfortable than young people (*n* = 14) in online groups and they tended to feel more positive about the online format overall [51]. In Israel, home-based online treatment during the pandemic brought mixed experiences, including patients’ reluctance to take part, a lack of access to technological devices, the impact of family and religious circumstances, whilst conversely, the virtual option was more positive for some patients and their parents [52].

### 3.4. Positives from the Pandemic

Some positives have arisen during the pandemic [22,23,27,28,30,48]. Greater connection with family, more time for self-care, and motivation to recover. Social media’s positive role in maintaining and facilitating social connections, accessing support amongst eating disorders communities, following others’ journeys to/of recovery. Family trying to bond, trying to challenge some eating disorder behaviours in a supportive way. Experiences of increased self-efficacy, seeking alternative practical coping strategies, gratitude for increased at home time. Lockdown as a catalyst for recovery, for some people. More time for healing and self-care, reaching out more for support and fewer worries related to shopping, events, meetings and work. Fewer stress provoking situations and less apprehension about appointments have also been noted.

Bodywhys also heard some positive reports from service users. Fewer demands outside the home meant that parents could focus their attention on the task at hand—establishing regular eating and supporting their person with an eating disorder. If everyone in the house was eating normally and regularly, it was easier for the person with the eating disorder to manage regular eating. People with eating disorders could experience and see the evidence that other people can eat regularly. Fewer social demands brought relief to some, especially around school and college demands and social interactions. In many cases, the lockdown brought opportunities for people within a family home to connect, to slow down, to start to understand each other better.

When an eating disorder becomes a part of someone’s life, a parent’s role can change into that of a carer, responding to the needs of the person who is unwell, whilst also managing the overall family situation. Previous Irish research has highlighted how feelings of isolation and helplessness can arise for both parents and young people affected by eating disorders [53]. Supporting parents and family members is therefore important to provide them with coping tools, a support structure, reduce their distress and offer them a space to feel heard. Providing support services such as the PiLaR programme online has meant that many more people can access the service, and they are not inhibited by travel distance, transport limitations, babysitters or schedules.

## 4. Conclusions

The Irish government announced a national lockdown in March 2020, in response to the COVID-19 pandemic. This led to several unprecedented changes in Irish society which was, in effect, shutdown, for example with a 2 km limitation on exercise outside of the home, along with the closures of schools and universities and asking people to work from home. People’s sense of routine and daily structure was significantly disrupted, social contact and the social fabric of society was altered in a way not previously seen.

This paper is not exhaustive, and the COVID-19 pandemic is ongoing. To date, there are a number of studies on the lived experience of eating disorders, and more are required to develop further insight, including the impact on carers. Currently, fewer studies have looked at the experience of carers and parents during the pandemic. Unsurprisingly, some of the concerns, such as accounts of restrictive eating and binge eating, have arising on the support services provided by Bodywhys are also reflected in the research literature.

Many challenges have arisen for people affected by eating disorders in Ireland. Risk factors have heightened, whilst protective factors have been less accessible. Sadly, as we have seen, when increased risk factors and reduced protective factors collide into each other, the emotional and personal dimensions to people’s lives have become very difficult and distressing. There was a sense that a ‘perfect storm’ had arisen, contributing to a very clear and immediate impact on people affected by eating disorders, but one which has also continued throughout the pandemic. A variety of situations have emerged: people relapsing, people being newly diagnosed during lockdown or the intensification of the lived experience for those with an existing issue. Due in part to the duration and intensity of the pandemic, and Ireland experiencing three lockdown periods, increased hospitalisations and worries for families have also become a concerning reality. Those working in the area, from clinicians to researchers, along with people affected by eating disorders, have had to adapt to new and unexpected circumstances, find alternative ways of working, adhere to social distancing (staying 2 m apart), and live with an illness that, by its nature, causes significant disruption in a person’s quality of life, relationships and health. Although the challenges to recovery are substantial, some positives have emerged, despite the intense nature of pandemic. Even with the incredibly difficult circumstances, it has also been a very productive time, services have adapted and expanded, views have been shared and voices heard, from clinicians and researchers to those with lived experience and family members. With knowledge of the challenges posed by the pandemic we can better ensure support services are equipped to deal with the many and varied difficulties that people affected by eating disorders and their carers face.

These challenges have not gone unnoticed in the Irish news media. One mother wrote an article [54] for an Irish national newspaper, The Irish Times, about how the lockdown had resulted in her daughter developing an eating disorder that caused her to be hospitalised.

‘My 11-year-old daughter is lying in a hospital bed struggling to eat one of seven meals the nurses will give her that day, while I try to get my head around how we ended up here. I ring my husband and relay everything I’ve learned in the past 24 h—our daughter is dangerously underweight and we have a lot of work ahead of us to help her get well.’

The lasting impact of these challenges on Ireland’s health services, organisations such as Bodywhys, and people who are experiencing eating disorders, remains to be seen. A limitation of this paper is that Bodywhys does not record or report information about how its services users are coping on an ongoing basis, or their health status, this falls outside the scope and role of the organisation. Each country applied its own set of social and movement restrictions, however it is beyond the scope of this paper to determine the impact on people affected by eating disorders on a country-by-country basis.

Despite its limitations, the online space has been a valuable mechanism for delivering support and more telehealth tools or services may emerge in time. Whenever the world enters a post-pandemic stage, the applicability, successes and strengths of online structures and strategies must not be used as justification for the dilution of effective specialised services. Accessing evidence-based treatment and supporting families remains key to helping a person to build a life after experiencing an eating disorder.

## Figures and Tables

**Figure 1 jcm-10-03385-f001:**
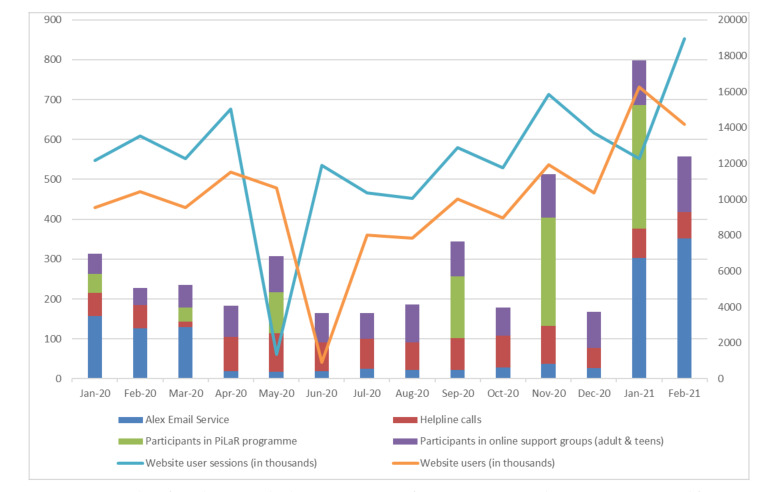
Number of people using Bodywhys’ support services from January 2020 to February 2021. Data sourced from Salesforce and Google Analytics.

**Figure 2 jcm-10-03385-f002:**
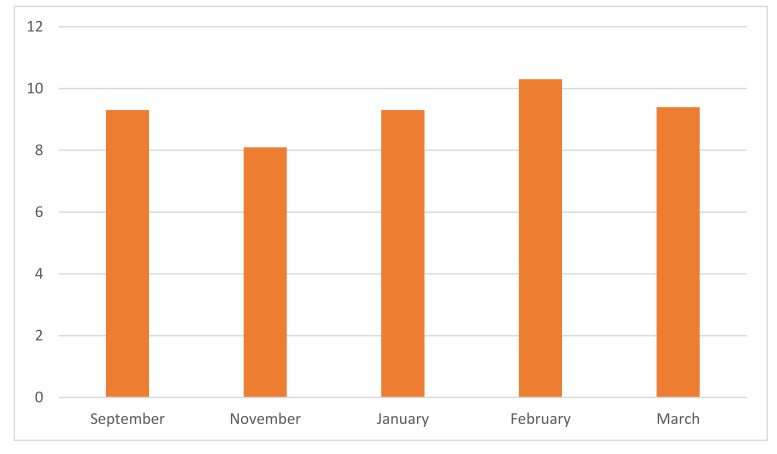
Fear of COVID-19 Scale (*n* = 627).

**Figure 3 jcm-10-03385-f003:**
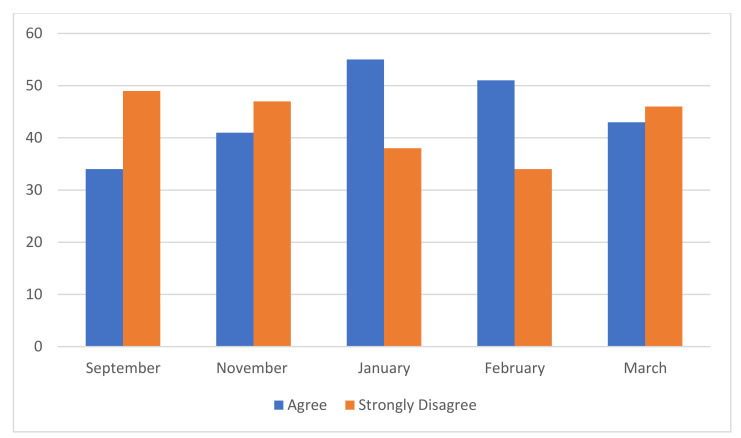
Impact of COVID-19 on carer’s ability to provide support: 5 rating scale to ‘COVID-19 has made it harder for me to provide support for the person with ED’. (*n* = 585).

**Figure 4 jcm-10-03385-f004:**
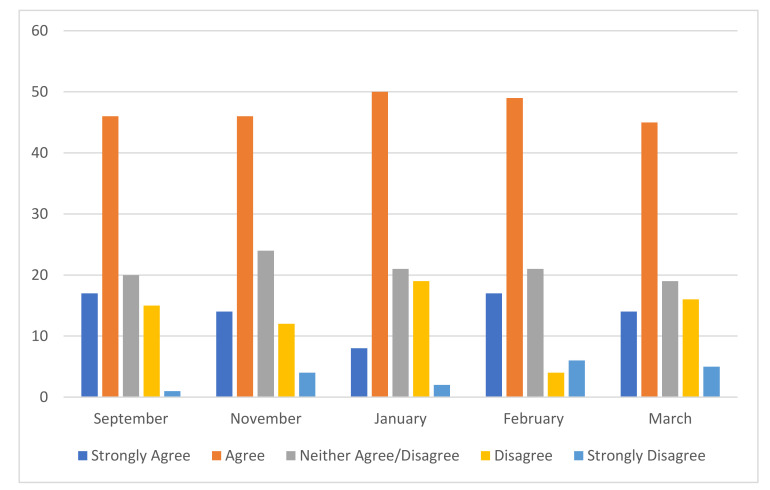
Impact on Mental Health of Carer: 5 rating scale to ‘My own mental health is being impacted by caring/supporting for someone with an ED’ (*n* = 585).

**Figure 5 jcm-10-03385-f005:**
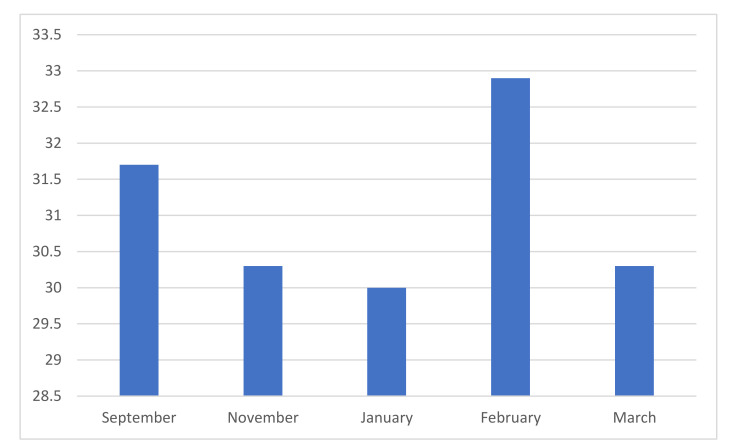
Accommodation and Enabling Scale (*n* = 627).

## Data Availability

The data presented in this study are available on request from the corresponding author. The data are not publicly available due to ethical issues and sensitive regarding operational issues.

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
