# Peer review of "Review of Ireland’s First Year of the COVID-19 Pandemic Impact on People Affected by Eating Disorders: ‘Behind Every Screen There Was a Family Supporting a Person with an Eating Disorder’"

_jcm, 2021, doi:10.3390/jcm10153385_

Round 1

Reviewer 1 Report

This is an interesting article mostly focused on the experience of the support network Bodywhys over the course of the year. There is a review of other data that is available. In general this is a well-written paper. I would suggest shrinking the section that describes the structure and function of Bodywhys- perhaps it could be included as a supplement.

Author Response

We thank the reviewers for their informative comments (29/05/21, 11/06/21) on our first draft.

Based on this feedback, we have:

  1. Added more information about Salesforce
  2. Added additional information about Bodywhys in the appendix
  3. Reviewed the query about the PiLaR data
  4. Updated some paragraphs and/or sentences to clarify phrasing and the fluidity of text
  5. Worked to strengthen the conclusion
  6. Checked spelling

Reviewer 2 Report

Parsons and colleagues examined changes in eating disorder service utilization in Ireland during the COVID-19 pandemic. Specifically, they examine Really unclear who the sample is 427 people included, but 629 completed data? In the paper’s introduction, the authors review Ireland’s response to the pandemic and outline the timing of restrictions and relaxation thereof. They present data already available about the numbers of individuals seeking treatment and being hospitalized for eating disorders pre- and during pandemic year. With the current paper, their focus is on the utilization of BodyWhys. Unfortunately, as someone who is less familiar with BodyWhys, I was not clear on the intersection between this service and other treatment offerings in Ireland, or of how representative individuals seeking care through BodyWhys were. Furthermore, in the Method, the authors describe extracting data from two sources—Salesforce, and surveys completed by folks involved in BodyWhys’ Pilar system. I am not familiar with Salesforce, and was not clear on how data would be merged or combined. I also was not clear who the participants in this study were. It sounds like there were 427 people included from BodyWhys’ Pilar, but that 629 completed data? Why the discrepancy? What data were gathered through Salesforce? It sounds as though much (all?) of what is provided for BodyWhys is currently online. Could it be that the significant uptick in service utilization was due to the move from in-person to virtual, rather than due to rise in EDs during the pandemic? Is the increase unique to eating disorders, or is a similar increase also seen in other psychiatric illnesses? Why would it be unique to eating disorders, and what does this mean? In Figures, it would be very helpful to include the possible ranges for the measures. Conclusions – it would be helpful if the authors could bring together the main points of the paper in synthesis. Why are these data important and what should the government (or treaters, or families) do differently going forward?

Author Response

We thank the reviewers for their informative comments (29/05/21, 11/06/21) on our first draft.

Based on this feedback, we have:

  1. Added more information about Salesforce
  2. Added additional information about Bodywhys in the appendix
  3. Reviewed the query about the PiLaR data
  4. Updated some paragraphs and/or sentences to clarify phrasing and the fluidity of text
  5. Worked to strengthen the conclusion
  6. Checked spelling

This manuscript is a resubmission of an earlier submission. The following is a list of the peer review reports and author responses from that submission.